# E3D: Enhancing Sparsely-Supervised 3D Object Detector with Large Multimodal Models

## Abstract

Recently, sparsely-supervised 3D object detection has gained great attention, achieving performance close to that of fully-supervised 3D objectors with only a few annotated instances. Nevertheless, these methods suffer challenges when the accurate labels are extremely limited. In this paper, we propose an **E**hanced **3**D object **D**etection strategy, termed **E3D**, explicitly utilizing the prior knowledge from Large Multimodal Models (LMMs) to enhance the feature discrimination capability of the 3D detector under sparse annotation settings. Specifically, we first develop a Confident Points Semantic Transfer (**CPST**) module that generates high-quality seed points through boundary-constrained center cluster selection. Based on these seed points, we introduce a Dynamic Cluster Pseudo-label Generation (**DCPG**) module that yields pseudo-supervision signals from the geometry shape of multi-scale neighbor points. Additionally, we design a Distribution Shape score (**DS score**) that chooses high-quality supervision signals for the initial training of the 3D detector. By utilizing E3D, existing leading sparsely-supervised CoIn++ is improved by an average of 11.63% under the annotation rate of 2%. Moreover, we have verified our E3D in the zero-shot setting, and the results demonstrate its performance exceeding that of the state-of-the-art methods. The code will be made publicly available.

## 1 Introduction

3D object detection, aiming at locating and classifying objects within 3D scenes, has garnered significant attention in autonomous driving (Wu et al., 2023; Deng et al., 2021; Liu et al., 2023c; Xia et al., 2023a; Huang et al., 2024). However, the performance of mainstream 3D detectors relies heavily on high-quality labels annotated by humans, which is not only time-consuming but also sensitive to the subjective impression of annotators.

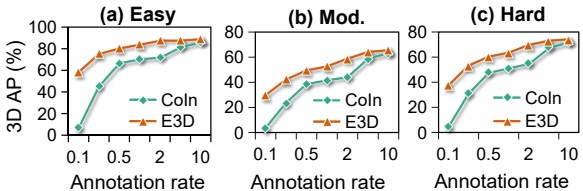

Figure 1: Performance comparison of the sparsely-supervised detector at various annotation rates. E3D indicates the CoIn initialized with the proposed E3D.

To minimize the dependence of 3D detectors on high-quality manual annotations, recent work has begun to focus on label-efficient training strategies (Liu et al., 2023a; Wang et al., 2021; Liu et al., 2022a; Xia et al., 2023b). To discover unlabeled instances, SS3D (Liu et al., 2022a) employs a self-training approach to iteratively optimize the detector trained on sparsely annotated data. CoIn (Xia et al., 2023b) introduces contrastive learning methods, enhancing the model's discriminative capability for various category features. However, existing strategies make 3D detectors struggle to extract sufficiently discriminative features from extremely limited annotations. Fig. 1 shows some examples where the state-of-the-art sparsely-supervised object detector, such as CoIn (Xia et al., 2023b), hardly maintains robust performance with a significant reduction in annotation rate.

Nowadays, with the successive emergence and widespread application of large language models (LLMs) such as BERT (Devlin et al., 2018) and GPT (Brown et al., 2020; Achiam et al., 2023) in natural language processing, research on large multimodal models (LMMs) is also gaining momentum (Radford et al., 2021; Li et al., 2022; Kirillov et al., 2023; Liu et al., 2023b). Benefiting from

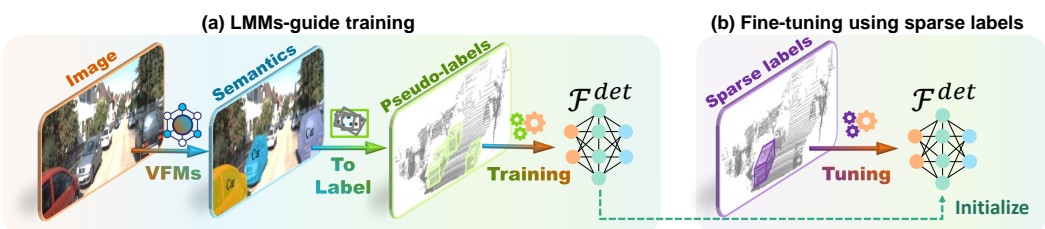

Figure 2: Illustration of E3D-assisted sparsely-supervised 3D object detection.

the outstanding performance of LMMs, the utilization of pre-trained LMMs has led to significant advancements in 2D vision tasks. Inspired by this, (Xue et al., 2023; Zhang et al., 2022; Zhu et al., 2023) transfers the image-text knowledge prior from 2D LMMs to 3D point clouds. However, these methods typically focus on the classification of individual instances, and there will be certain limitations when applying them directly to outdoor 3D object detection. Despite this, these attempts that transfer the priors from 2D LMMs to 3D point clouds still provide a new perspective for solving the problem of sparsely-supervised 3D object detection.

Motivated by the methods above, we proposed a two-stage training strategy, termed **E3D**, enhancing sparsely-supervised 3D object detection based on LMMs. As shown in Fig. 2, we first employ LMMs to extract semantics from 2D images and explicitly transfer them to 3D point clouds, generating pseudo-labels for the first stage of detector training. In the second stage, we fine-tune the trained model using a small amount of accurate labels. Specially, E3D is built upon two basic questions: (1) *How to accurately transfer semantic information obtained from LMMs in 2D images to 3D point clouds*. Due to the absence of inherent depth information in images, directly transferring image semantics onto point clouds may result in noisy semantics at the edge of the instance. (2) *How to efficiently utilize the LMMs-extracted semantics to enhance sparsely-supervised 3D object detection*. Based on the obtained semantics, directly fitting pseudo-labels may result in incomplete foreground bounding boxes.

Based on the questions mentioned above, our E3D first designs a Confident Points Semantic Transfer (**CPST**) module, obtaining 3D seed points through boundary-constrained center cluster selection. These seed points focus on central foreground semantic masks generated by LMMs. Inspired by unsupervised algorithms (Zhang et al., 2023; Wu et al., 2024), we can utilize these seed points to generate bounding box pseudo-labels. In this case, we introduce a Dynamic Cluster Pseudo-label Generation (**DCPG**) module and Distribution Shape score (**DS score**) to discover high-quality pseudo-labels with complete foreground information from seed points. As shown in Fig. 2, we utilize the generated pseudo-labels to train the 3D object detector for the first stage. After training, the 3D detector has learned a certain of feature discrimination capability from the 2D images. Subsequently, we fine-tune the 3D detector with sparse accurate labels, and in conjunction with current label-efficient methods, it demonstrates relatively high detection capabilities even under extremely low labeling scenarios. The contributions of this paper can be summarized as follows:

- We propose an **E**hanced **3**D object **D**etection strategy (**E3D**), utilizing 2D image and LMMs to boost the feature discrimination capability of 3D detector under sparsely-supervised situations. E3D provides an initial detector with a stronger feature extraction capability, enabling stable detection performance despite continuous reduction in annotated instances.

- We propose a Confident Points Semantic Transfer (**CPST**) module, which leverages LMMs to obtain accurate semantic seed points. Subsequently, we propose Dynamic Cluster Pseudo-label Generation (**DCPG**) module and Distribution Shape score (**DS score**) for high-quality pseudo-label generation based on the seed points, which will be applied in the first-stage training process.

- Experiment results on the KITTI dataset show that E3D substantially enhances the performance of leading sparsely-supervised 3D object detectors. By utilizing E3D, CoIn is improved by 36.92% and 14.89% under the annotation rate of 0.1% and 2%. Moreover, without fine-tuning on labeled data, our E3D has shown superior performance compared to zero-shot methods, demonstrating the effectiveness of E3D-initialized detector.

## 2 RELATED WORK

### 2.1 LABEL-EFFICIENT 3D OBJECT DETECTION

Recently, label-efficient 3D object detection methods have begun to be explored in responding to the challenge of extremely low annotation volumes. Generally, these label-efficient methods can be categorized into semi-supervised (Wang et al., 2021; Zhao et al., 2020; Park et al., 2022; Liu et al., 2023a), and weakly-supervised (Meng et al., 2021; Liu et al., 2022b) approaches according to the difference in quantity and supervision form. To maintain accuracy while reducing the annotations, SS3D (Liu et al., 2022a) introduces the concept of sparse supervision, annotating only one complete 3D object per frame. Based on SS3D, CoIn (Xia et al., 2023b) adopts a contrastive instance feature mining strategy, enabling the extraction of feature-level pseudo-labels from a significantly reduced amount of annotated data. However, the performance of existing methods remains constrained due to the insufficient feature discriminability of the initial detector, which may affect subsequent training under very few annotations. This work aims to develop a two-stage strategy, enabling the 3D detector to maintain robust feature representation capabilities despite having lower instance annotations.

### 2.2 LARGE MULTIMODAL MODELS IN 3D

As the outstanding performance achieved by LMMs in 2D tasks (Radford et al., 2021; Kirillov et al., 2023; Liu et al., 2024; Peebles & Xie, 2023; Bai et al., 2023), some studies are beginning to explore their application in the 3D domain. Inspired by CLIP (Radford et al., 2021), ULIP (Xue et al., 2023) enhances the 3D understanding capability by transferring knowledge from 2D LMM to 3D encoder through contrastive learning methods. Similar works are (Zhang et al., 2022; Zhu et al., 2023). In the outdoor scenario, SAM3D (Zhang et al., 2024) employs SAM to segment BEV images of point clouds and fit bounding boxes based on the segmentation masks to obtain detection results. CLIP2Scene (Chen et al., 2023b) establishes the connection between point clouds and text by using images as an intermediate modality, enhancing the 3D model's semantic understanding of the scene with the prior knowledge of CLIP. Differing from previous approaches, our E3D explicitly transfers semantic masks obtained from LMMs onto point clouds to generate high-quality pseudo-labels for the first-stage training of the 3D detector.

### 2.3 MULTIMODAL REPRESENTATION LEARNING

Recently, the utilization of multimodal from 2D images and 3D point clouds to enhance 3D object detectors has been gradually gaining the attention of the community (Liu et al., 2023c; Wang et al., 2023; Wu et al., 2023; Song et al., 2024; Xie et al., 2023). However, these works mainly focus on investigating the image-point cloud fusion strategy, neglecting the utilization of images to explore label-efficient 3D detection. To reduce the required annotations, some methods transfer 2D image information into 3D point clouds to generate pseudo-labels (Yang et al., 2024). However, semantic ambiguity may occur at the edge of the object due to the 2D-3D calibration error. MixSup (Yang et al., 2024) proposed a connected components labeling strategy, addressing this issue with the spatial separability property inherent to point clouds. SAL employ (Yang et al., 2024) employ a density-based clustering to refine imperfect projection issues. Compared with these methods, our E3D provides a simple but efficient way to reduce semantic noise caused by projection errors.

## 3 METHODS

### 3.1 OVERVIEW

This paper introduces an **E**hanced **3D** object **D**etection strategy (**E3D**), explicitly utilizing the prior knowledge from LMMs to boost the sparsely-supervised 3D detectors. As shown in Fig. 3, E3D consists of three primary components: (1) **Confident Points Semantic Transfer (CPST)** module, which acquires high-quality seed points with the boundary-constrained center cluster selection. (2) **Dynamic Cluster Pseudo-label Generation (DCPG)** module, which dynamically generates pseudo-label proposals based on the geometric shapes within the multi-scale neighborhood of these seed points. (3) **Distribution Shape score (DS score)**, which employs unsupervised priors as the criterion for evaluating the quality of pseudo-label proposals, and we subsequently apply non-

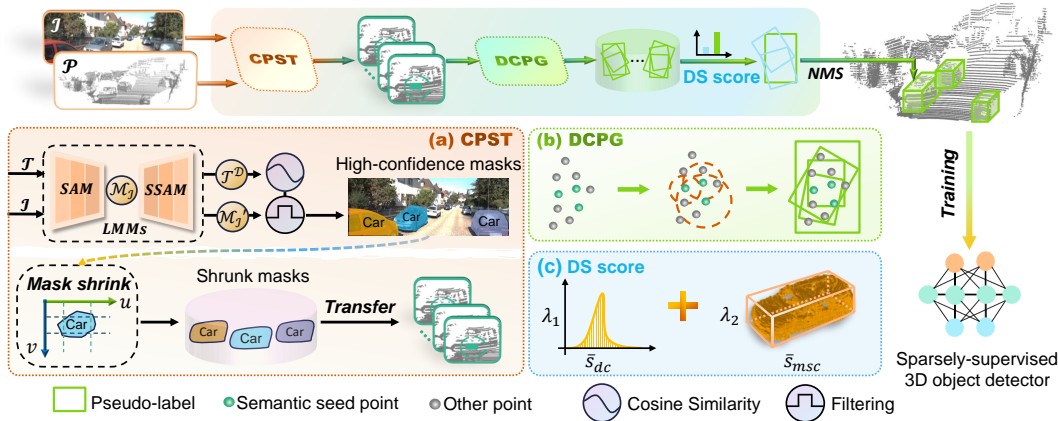

Figure 3: The overview of our E3D, including (a) CSPT finds semantic seed points through high-confidence semantic masks transfer, (b) DCPG dynamically clusters neighbor points of seed points to fit pseudo-label proposals, and (c) DS score to evaluate the quality of generated pseudo-label proposals, serving as a scoring metric to NMS to suppress low-quality proposals.

maximum suppression (NMS) (Neubeck & Van Gool, 2006) to retain high-quality pseudo-labels further. We then follow the training strategy of CoIn (Xia et al., 2023b) to train an initial 3D detector with enhanced discriminative capacity. We detail our E3D framework as follows.

## 3.2 CONFIDENT POINTS SEMANTIC TRANSFER MODULE

Encouraged by the development of LMMs, we first utilize LMMs to extract semantic information from 2D images explicitly. Meanwhile, by integrating the projection relationship matrix between images and point clouds, it is quite straightforward to transfer semantic information onto point clouds. However, as shown in Fig. 4, there is significant noise in the edge points of instances during the process transfer. To prevent the incorporation of noise during the transfer of semantic information, we use the boundary-constrained *mask shrink* operation, followed by the coordinate system trans-

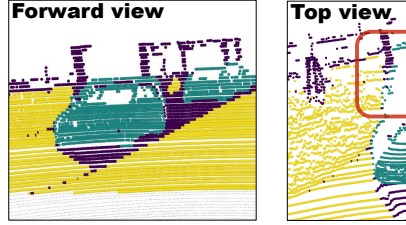

Figure 4: Semantic transfer noise. Semantic belonging to the same objects may be assigned to different instances.

formation, to obtain accurate semantic seed points. CSPT is illustrated in Fig. 3(a); we have divided it into two parts as **LMMs-guide semantic extraction** and **confident points filtering**.

**LMMs-guide semantic extraction.** The goal of LMMs is to generate high-quality foreground semantic masks. Specifically, we take as input an image $\mathcal{I} \in \mathbb{R}^{3 \times H \times W}$ and $C$ text prompt $\mathcal{T}^{\mathcal{C}} = \left\{ t_1^{\mathcal{C}}, t_2^{\mathcal{C}}, ..., t_C^{\mathcal{C}} \right\}$, where $H$ and $W$ denote the height and width of the image. We first utilize FastSAM (Zhao et al., 2023) to perform speed-efficient segmentation as

$$\mathcal{M}_{\mathcal{I}} = \text{SAM}(\mathcal{I}), \tag{1}$$

where $\mathcal{M}_{\mathcal{I}} \in \mathbb{R}^{M \times H \times W}$ denotes the $M$ class-agnostic masks extracted from $\mathcal{I}$, and $\text{SAM}(\cdot)$ indicates the FastSAM model. We then utilize $\mathcal{M}_{\mathcal{I}}$ as the mask prompts and feed them, along with image $\mathcal{I}$, into SemanticSAM (Chen et al., 2023a) model, which is except to output the descriptions $\mathcal{T}^{\mathcal{D}} = \left\{ t_1^{\mathcal{D}}, t_2^{\mathcal{D}}, ..., t_M^{\mathcal{D}} \right\}$ for each mask in $\mathcal{M}_{\mathcal{I}}$. Specifically:

$$\mathcal{T}^{\mathcal{D}} = \text{SSAM}(\mathcal{I}, \mathcal{M}_{\mathcal{I}}), \tag{2}$$

where $\text{SSAM}(\cdot)$ refers to the SemanticSAM model. Generally, the elements in $\mathcal{T}^{\mathcal{C}}$ represent the categories of interest. Therefore, we calculate the cosine similarity between $\mathcal{T}^{\mathcal{C}}$ and $\mathcal{T}^{\mathcal{D}}$ to filter out uninteresting background masks, thereby obtaining the foreground masks $\mathcal{M}'_{\mathcal{I}}$.

**Confident points filtering.** As fuzziness of boundaries results from depth occlusions in images and calibration inaccuracies, we opt to constrain the boundary of foreground masks before 2D-3D transfer, retaining only its central portion. Specifically, for each foreground mask, we denote its maximum and minimum values in the pixel coordinate system $(u, v)$ as $(u_{min}, u_{max}, v_{min}, v_{max})$. We perform *mask shrink* to constraint boundary range of $\mathcal{M}'_{\mathcal{I}}$ to obtain $\hat{\mathcal{M}}'_{\mathcal{I}}$:

$$
\begin{aligned}
u \in [u_{min} + \frac{1}{2}(1 - \gamma)(u_{max} - u_{min}),\ u_{min} + \frac{1}{2}(1 + \gamma)(u_{max} - u_{min})], \\
v \in [v_{min} + \frac{1}{2}(1 - \gamma)(v_{max} - v_{min}),\ v_{min} + \frac{1}{2}(1 + \gamma)(v_{max} - v_{min})],
\end{aligned}
\tag{3}
$$

where $\gamma$ denotes the shrink factor. With this constraint, we obtain the shrunk masks that filter out the semantically ambiguous regions, ensuring the accuracy of foreground semantic information transferred onto the point clouds. Following (Vora et al., 2020), we transfer the semantic mask from the image onto the point clouds to obtain semantic seed points using the camera's intrinsic and extrinsic parameter matrices. It is worth noting that we adopt the approach of explicitly transferring the shrunk masks onto the point cloud rather than implicitly embedding unprocessed semantic masks into the point cloud's features as (Vora et al., 2020). This approach helps avoid potential semantic feature confusion between different modalities arising from sparse annotations.

### 3.3 DYNAMIC CLUSTER PSEUDO-LABEL GENERATION MODULE

With the assistance of CPST, we explicitly obtain the semantic seed points from transformation $\hat{\mathcal{M}}'_{\mathcal{I}}$ of the foreground mask. Given $\mathcal{P} \in \mathbb{R}^{N \times 3} = \{p_1, p_2, ..., p_N\}$ as LiDAR points, we define the seed points covered by $\hat{\mathcal{M}}'_{\mathcal{I}}$ as $\mathcal{P}_T = \{p_t\}$, $\mathcal{P}_T \subset \mathcal{P}$. It is crucial for the 3D detection task that obtain complete bounding boxes from these seed points.

By referring to traditional unsupervised pseudo-label generation methods (Zhang et al., 2023), we produce a large number of pseudo-labels and then use the positional constraints of the seed points to retain the more promising pseudo-labels as supervisory signals. However, existing unsupervised bounding box fitting approaches (Zhang et al., 2023; Wu et al., 2024) usu-

---

**Algorithm 1** Dynamic cluster pseudo-label generation

**Input:** LiDAR points $\mathcal{P}$, the $k$-th seed points $\mathcal{P}_T^{(k)} = \left\{ p_t^{(k)} \right\}$, initial radius $r_{initial}$;

**Output:** Pseudo-label proposal set $\hat{\mathcal{B}}^{(k)}$

1: $N^{(k)} = \mathcal{P}_T^{(k)}.\text{shape}[0]$
2: $\mathcal{P}_{gr} \leftarrow \text{GroundRemove}(\mathcal{P})$
3: $\hat{\mathcal{B}}^{(k)} = [\,]$
4: **for** $t = 1, 2, ..., N^{(k)}$ **do**
5:     $p_t = \mathcal{P}_T^{(k)}[t]$
6:     $r \leftarrow \text{update}(t, r_{initial})$
7:     $\hat{\mathbf{b}} \leftarrow \text{BoxFit}(\text{DBSCAN}(\mathcal{P}_{gr}, p_t, r))$
8:     $\hat{\mathcal{B}}^{(k)}.\text{append}(\hat{\mathbf{b}})$
9: **end for**

---

ally take a fixed constant as cluster radius, leading to the problem of inadequate foreground or excessive background noise for the generated bounding boxes. In this case, we propose a dynamic cluster pseudo-label generation (DCPG) module. This module utilizes the geometry shape of the seed points' multi-scale neighborhood to capture complete foreground information while minimizing background interference. It dynamically generates pseudo-label proposals.

Specifically, we denote $\mathcal{P}_T^{(k)}$ as the $k$-th instance in a point cloud frame and utilize DCPG dynamically generates a clustering radius $r$ for the $t$-th seed point $p_t^{(k)}$. We define the updating function for the dynamic radius as

$$
\text{update}(t, r_{initial}) = r_{initial} \cdot \frac{t}{N^{(k)}} + \delta, t = 1, 2, ..., N^{(k)},
\tag{4}
$$

where $r_{initial}$ is a hyper-parameter set based on empirical experience, $\delta$ denotes the adjustment factor to avoid $r$ too small, and $N^{(k)}$ is the number of seed points in the current instance. By applying Eq. 4, we dynamically update the radius $r$, $r \in (\delta, r_{initial} + \delta]$, during point clustering, thereby obtaining foreground clusters with multi-scale receptive fields. Following (Zhang et al., 2023). We utilize the radius calculated from Eq. 4 as the clustering radius for DBSCAN (Ester et al., 1996) and employ (Zhang et al., 2017) to fit the bounding box for each foreground cluster. Algorithm 1 summarizes Our DCPG.

## 3.4 DISTRIBUTION SHAPE SCORE

While DCPG has the capacity for high-quality pseudo-label generation, it unavoidably produces an amount of low-quality pseudo-label proposals. The shapes of these proposals and the extent of foreground completeness contained within the proposals usually deviate significantly from reality. Traditional detection methods typically compute the Intersection over Union (IoU) score between predicted bounding boxes and ground-truth (GT) boxes and then employ the NMS (Neubeck & Van Gool, 2006) to suppress these low-quality proposals. However, lacking GT makes it challenging to directly apply NMS using IoU as the evaluation criterion within our E3D framework. In this case, we propose a distribution shape score (DS score) to assess the quality of the pseudo-labels using unsupervised prior knowledge. We divided the DS score into two parts: **distribution constraint score** and **meta-shape constraint score**.

**Distribution constraint score.** Inspired by (Luo et al., 2024), within a high-quality pseudo-label proposal $\hat{b}$, the distances from its interior points $p_{i,i=1,...,n}$ to its boundary roughly follow a Gaussian distribution $\mathcal{N}(\mu, \sigma)$, where $\mu = 0.8$ and $\sigma = 0.2$, respectively. In other words, we denote random variable $D = \{d_1, ..., d_n\}$ as the distance between $p_i$ and the box boundary of $\hat{b}$, and $D \sim \mathcal{N}(0.8, 0.2)$. Based on this prior, we assign a distribution constraint score to the pseudo-label proposal $\hat{b}$ by calculating the similarity between the random variable $D$ corresponding to each $\hat{b}$ and the normal distribution $\mathcal{N}$. Specifically:

$$s_{dc}(\hat{b}) = \frac{1}{|\mathcal{P}_{fg}|} \sum_{p_i \in \mathcal{P}_{fg}} \log(\mathcal{N}(D|\mu, \sigma)), \tag{5}$$

where $\log(\cdot)$ denotes the logarithm function, $\mathcal{P}_{fg}$ is the foreground points within $\hat{b}$, and $|\mathcal{P}_{fg}|$ is the number of points in $\mathcal{P}_{fg}$.

**Meta-shape constraint score.** In addition, the shape of a high-quality pseudo-label is expected to be consistent with its template in the real world, which we define as the meta instance, corresponding to its category (Wu et al., 2024). For class $c$, we denote $\mathcal{B}_c \in \{l_c, w_c, h_c\}$ as the shape of its meta instance, where $l_c$, $w_c$ and $h_c$ are the normalized length, width and height, respectively. we followed this shape prior to constructing the class-aware meta-shape constraint score $s_{msc}(\hat{b})$ as

$$s_{msc}(\hat{b}) = 1 - \Phi_{KL}(\mathcal{B}_c || \hat{\mathcal{B}}_{\hat{b}}), \tag{6}$$

where $\Phi_{KL}(\cdot)$ denotes the normalized KL divergence function, and $\hat{\mathcal{B}}_{\hat{b}} \in \{l_{\hat{b}}, w_{\hat{b}}, h_{\hat{b}}\}$ indicates the normalized shape of the pseudo-label proposal. The purpose of this operation is to suppress the low-quality proposals whose shape deviates significantly from the meta instance. By combining the distribution constraint score and the meta-shape constraint score, we can obtain the DS score as

$$\text{DS}(\hat{b}) = \lambda_1 \overline{s}_{dc}(\hat{b}) + \lambda_2 \overline{s}_{msc}(\hat{b}), \tag{7}$$

where $\lambda_1$ and $\lambda_2$ denote weight adjustment factor. Notably, to unify the dimension, we normalized the two constraint scores before combining them, resulting in $\overline{s}_{dc}$ and $\overline{s}_{msc}$. We then employ the DS score as a substitution for the confidence score in NMS to suppress the low-quality pseudo-labels. We utilize the obtained pseudo-labels in conjunction with CoIn (Xia et al., 2023b) for the first phase of training. Subsequently, we fine-tune the trained detector with a small amount of accurate labels to boost the performance of the 3D detector.

## 4 EXPERIMENTS

**Dataset and metrics.** As one of the large-scale benchmark datasets in autonomous driving, the KITTI (Geiger et al., 2012) dataset has been widely used in 3D object detection. During the first training stage, we did not use any ground truth for training. Instead, we relied solely on the semantic information provided by LMMs to generate pseudo-labels. In the fine-tuning stage, we followed the recent works (Deng et al., 2021; Xia et al., 2023b) to split the training set (contains 7,481 scenes) into a *train* split (contains 3,712 scenes) and a *val* split (contains 3,769 scenes). We then randomly select 10% of the scenes from the *train* split and retain only one instance annotation per scene. In this case, we can obtain a *limited* split, which merely takes 2% of instance annotations compared with the origin *train* split. To guarantee a fair comparison, we present the results with the primary official evaluation metric: 3D Average Precision (AP) across 40 recall thresholds (R40).

**Implementation Details.**   **Pseudo-labels Generation:** We directly employed the model parameters provided by FastSAM (Zhao et al., 2023) and SemanticSAM (Chen et al., 2023a) implementations for inference, without additional supervisory signals for fine-tuning. To achieve accurate segmentation results, we set a higher segmentation threshold of 0.7 during the FastSAM inference process. To mitigate computational demands, we opted to generate pseudo-labels within a confined spatial domain of the semantically relevant points, specifically within an 8-meter radius. We set mask shrink factor $\gamma$ to 0.3, initial cluster radius $r_{initial}$ to 1, adjustment factor $\delta$ to 0.1. We utilize unsupervised priors to filter out pseudo-labels that are evidently inconsistent with the intuitive expectations and set the weight adjustment factor of DS score $\lambda_1$ and $\lambda_2$ as 0.5, 0.5. **Detector Training:** We conduct all experiments with a batch size of 8 and a learning rate of 0.003 on 4 RTX 3090 GPUs. Following previous sparsely-supervised 3D object detection methods (Xia et al., 2023b; Liu et al., 2022a), we choose three different classical detectors (Yin et al., 2021; Deng et al., 2021; Wu et al., 2022) as our architecture. And we employ the OpenPCDet (Team et al., 2020) to conduct our experiments. In the first training stage, we employ CoIn (Xia et al., 2023b) to train an initial detector with the generated pseudo-labels. Then, we use *limited* split to fine-tune the detector.

**Baselines.**   To thoroughly validate the effectiveness of the proposed E3D, we select the state-of-the-art (SoTA) sparsely-supervised methods Xia et al. (2023b) as the primary baseline for comparison. We compare the proposed E3D approach with the baseline under conventional sparse settings with 2% annotation cost. We also compared with cross-modal weakly-supervised methods (Qin et al., 2020; Liu et al., 2022b), which also incorporate visual models to extract semantic information to enhance the performance of weakly-supervised detectors. Furthermore, we establish baselines under progressively reduced annotation costs to evaluate the sensitivity to annotation costs.

## 4.1 COMPARISON WITH SoTA METHODS

Table 1: Comparsion with SoTA sparsely-supervised methods on KITTI $val$ split. All methods are based on VoxelRCNN, and we report the 3D AP results of full cost (100%) and limited cost (20%, 2%). The best sparsely-supervised methods are highlighted in **bold**.

| Setting | Cost | Method | Car | | | Ped | | | Cyc | | |
|---|---|---|---|---|---|---|---|---|---|---|---|
| | | | Easy | Mod. | Hard | Easy | Mod. | Hard | Easy | Mod. | Hard |
| *Fully-sup.* | *100%* | *VoxelRCNN* | *92.3* | *84.9* | *82.6* | *69.6* | *63.0* | *58.6* | *88.7* | *72.5* | *68.2* |
| Sparsely-sup. | 20% | SS3D | 89.3 | 84.2 | 78.2 | - | - | - | - | - | - |
| | 2% | VoxelRCNN | 70.5 | 54.9 | 44.8 | 42.6 | 38.5 | 32.1 | 73.3 | 47.8 | 43.2 |
| | 2% | CoIn | 89.1 | 70.2 | 55.6 | 50.8 | 45.2 | 39.6 | 80.2 | 52.3 | 48.6 |
| | 2% | CoIn++ | **92.0** | 79.5 | 71.5 | 46.7 | 36.1 | 31.2 | 82.0 | 58.4 | 54.6 |
| | 2% | CoIn++ with E3D | 91.3 | **80.5** | **74.0** | **67.4** | **58.7** | **50.9** | **92.5** | **73.1** | **68.3** |

**Comparison with sparsely-supervised methods.**   We integrate our proposed E3D into the SoTA sparsely-supervised 3D detection algorithm, CoIn++ Xia et al. (2023b). For a fair comparison, all detectors employ the VoxelRCNN Deng et al. (2021) as the base architecture. As illustrated in Tab. 1, E3D significantly improves the detection performance of CoIn++. Concurrently, we observe a slight decrease in precision for the 'Easy' car category with our E3D-initialized model. This could arise because our initial pseudo-labels are inferred based on the geometric shape of the objects, which may differ from the conventions of manual annotation. When the point cloud structure of an instance is relatively intact, such discrepancies can lead to noticeable differences in the size of the annotated bounding boxes.

**Comparison with fully-supervised methods.**   For a fair comparison, consistent with the approach of CoIn (Xia et al., 2023b), we select CenterPoint (Yin et al., 2021), VoxelRCNN (Deng et al., 2021), and CasA (Wu et al., 2022) as our baseline detectors, representing three distinct types of detection algorithms. We initialize the 3D detector using our E3D, followed by fine-tuning with the *limited* split. As shown in Tab. 2, due to limited annotations, it is difficult for the detectors designed under a fully supervised setting to achieve good detection results. Despite the effectiveness of CoIn in improving this situation, the results achieved are still unsatisfactory for single-stage detection algorithms with relatively simple structures. Our designed strategy, E3D, significantly reduces this discrepancy, enabling detectors to achieve similar results.

Table 2: Comparison with different fully-supervised methods. Sparse label refers to the use of *limited* split (2% annotation cost). The 3D object detection and BEV detection benchmark are evaluated by mean average precision with R40, under IoU thresholds 0.7.

| Stage | Label | Method | Car-3D @IoU 0.7 | | | Car-BEV @IoU 0.7 | | |
|---|---|---|---|---|---|---|---|---|
| | | | Easy | Mod | Hard | Easy | Mod | Hard |
| Single-stage | Fully | 1.CenterPoint | 89.07 | 80.50 | 76.49 | 92.98 | 89.01 | 87.50 |
| | Sparse | 2.CenterPoint | 49.69 | 31.55 | 25.91 | 56.78 | 42.50 | 34.14 |
| | Sparse | 3. CoIn(CenterPoint-based) | 72.03 | 54.82 | 43.77 | 87.20 | 73.54 | 66.03 |
| | Sparse | 4. 3 with E3D(CenterPoint-based) | 87.44 | 69.24 | 58.61 | 92.72 | 80.00 | 69.01 |
| | - | 5. *Improvements 4→1* | -1.63 | -11.26 | -17.88 | -0.26 | -9.01 | -18.49 |
| | - | 6. *Improvements 4→3* | 16.41 | 14.42 | 14.84 | 5.52 | 7.54 | 2.98 |
| Two-stage | Fully | 1.Voxel-RCNN | 92.38 | 85.29 | 82.86 | 95.52 | 91.25 | 88.99 |
| | Sparse | 2.Voxel-RCNN | 70.52 | 54.97 | 44.82 | 83.67 | 71.14 | 57.71 |
| | Sparse | 3. CoIn(Voxel-RCNN-based) | 84.56 | 68.47 | 58.02 | 92.31 | 81.01 | 70.24 |
| | Sparse | 4. 3 with E3D(Voxel-RCNN-based) | 91.37 | 74.89 | 63.84 | 95.41 | 85.27 | 74.57 |
| | - | 5. *Improvements 4 →1* | -1.01 | -10.4 | -19.02 | -0.11 | -5.98 | -14.42 |
| | - | 6. *Improvements 4→3* | 6.81 | 6.42 | 5.82 | 3.1 | 4.26 | 4.33 |
| Multi-stage | Fully | 1.CasA | 93.08 | 86.33 | 81.86 | 93.93 | 90.20 | 87.72 |
| | Sparse | 2.CasA | 74.18 | 57.37 | 45.05 | 85.90 | 73.21 | 57.23 |
| | Sparse | 3.CoIn(CasA-based) | 89.17 | 75.32 | 62.98 | 95.99 | 85.02 | 72.47 |
| | Sparse | 4. 3 with E3D(CasA-based) | 91.12 | 75.94 | 66.46 | 96.55 | 85.65 | 76.31 |
| | - | 5. *Improvements 4→1* | -1.96 | -10.39 | -15.4 | +2.62 | -4.55 | -11.41 |
| | - | 6. *Improvements 4→3* | 1.95 | 0.62 | 3.48 | 0.56 | 0.63 | 3.84 |

**Comparison with cross-modal weakly-supervised methods.** We also compare our E3D (CasA-based) with the SoTA cross-modal weakly-supervised 3D detection methods under the zero-shot setting. In VS3D (Qin et al., 2020) and WS3DPR (Liu et al., 2022b), they both use the pre-training semantic-processing model to support the semantic information to the detector. As shown in Tab. 3, compared with previous methods, by introducing semantic information from large multimodal models and then utilizing the designed pseudo-label generation module, our detection results are leading by a wide margin.

Table 3: Comparison with cross-modal weakly-supervised methods. We report the results with 40 recall positions, under 0.5 and 0.7 IoU thresholds.

| Method | Car-3D @IOU 0.5/0.7 | | |
|---|---|---|---|
| | Easy | Mod | Hard |
| VS3D | 31.09/9.09 | 37.36/5.73 | 40.32/5.03 |
| WS3DPR | -/60.01 | -/44.48 | -/36.93 |
| Ours E3D | 93.75/69.71 | 76.36/48.65 | 71.01/40.53 |

**Comparison with different annotation rates.** To explore the influence of our proposed E3D on the sparsely-supervised algorithm, we conducted a group of comparative experiments under different annotation rates. Tab. 4 provides the variation in performance as annotation rates ranging from 10% to 0.1%. Following the previous method (Liu et al., 2022b), we select a two-stage detector as a base detector for fair comparison. The experimental results indicate that the original sparsely-supervised 3D detector can significantly enhance performance upon integrating the proposed E3D. For example, at a 2% labeling rate, the CoIn integrated with E3D improved 3D AP by 15.41%, 14.42%, and 14.84% on easy, moderate, and hard difficulty levels, respectively. Also, this result represents an average improvement of 14.89% over the original detector. Be-

Table 4: Comparison with different annotation rates (10% → 0.1%). We report the results with 40 recall positions, under 0.7 IoU threshold.

| Annotation Rate | Method | Car-3D @IoU 0.7 | | |
|---|---|---|---|---|
| | | Easy | Mod | Hard |
| 100% | CenterPoint | 89.07 | 80.50 | 76.49 |
| 10% | CoIn | 85.95 | 71.80 | 62.64 |
| | + E3D | 88.84 | 73.56 | 65.17 |
| 5% | CoIn | 81.64 | 67.48 | 58.32 |
| | + E3D | 87.52 | 72.42 | 63.87 |
| 2% | CoIn | 72.03 | 54.82 | 43.77 |
| | + E3D | 87.44 | 69.24 | 58.61 |
| 1% | CoIn | 70.39 | 51.31 | 41.31 |
| | + E3D | 83.79 | 63.16 | 52.50 |
| 0.5% | CoIn | 66.77 | 47.68 | 38.38 |
| | + E3D | 80.36 | 59.99 | 49.44 |
| 0.2% | CoIn | 45.47 | 31.20 | 23.52 |
| | + E3D | 75.30 | 52.99 | 42.14 |
| 0.1% | CoIn | 6.84 | 4.65 | 3.61 |
| | + E3D | 58.57 | 37.41 | 29.88 |

sides, our E3D significantly boosts the sparsely-supervised 3D detector's performance even at very low annotation rates, which achieves the 41.95% (36.92% higher than CoIn) average AP across dif-

ferent difficult levels under the annotation rate of 0.1%. The experimental results indicate that the performance of the original sparsely-supervised 3D detector can improve significantly after loading the E3D-initialized model, even at low annotation rates.

## 4.2 ABLATION STUDY

**Effectiveness of mask shrink, DCPG, and DS Score.** To rapidly verify the effectiveness of the proposed modules, we conducted ablation studies based on CenterPoint (Yin et al., 2021) and recorded the results in Tab. 5. The results presented in the first and second rows illustrate that the precision of pseudo-labels, as augmented by the multi-scale neighborhood clustering mechanism within DCPG, can substantially amplify the detection capabili-

Table 5: Effects of the different components of E3D. We report the mAP with R40, under IoU threshold 0.7.

| Mask shrink | DCPG | DS score | 3D-Car AP@IoU 0.7 | | |
|:---:|:---:|:---:|:---:|:---:|:---:|
| | | | Easy | Mod. | Hard |
| ✓ | | | 35.10 | 23.75 | 19.52 |
| ✓ | ✓ | | 40.56 | 28.15 | 22.40 |
| | ✓ | ✓ | 47.23 | 33.40 | 27.13 |
| ✓ | ✓ | ✓ | 52.56 | 38.00 | 31.52 |

ties of the 3D detector. This may be attributed to incorporating more comprehensive foreground information in the generated pseudo-labels, which has enhanced the model's feature discrimination capability. The comparison between the third and fourth rows of the table demonstrates that the mask shrink operation is necessary for handling semantic noise at the instance edges. Moreover, the results from the second and fourth rows indicate that using the DS score for filtering out low-quality labels can significantly enhance the precision of the detector. When combined, the three modules facilitate the most accurate information transfer and pseudo-label generation, enabling the 3D detector obtained from the first-stage training with more robust performance, promoting subsequent fine-tuning with accurate labels.

**The comparison of recall on different IoU thresholds.** To verify the positive impact of the proposed E3D on recognition, we evaluated the recall rates under different IoU thresholds. As depicted in Tab. 6, the E3D model consistently elevates recall rates across the different IoU thresholds, demonstrating a stable improvement. Since the geometric information we provide is derived from rule-based

Table 6: The comparison of Recall on different IoU thresholds (0.3, 0.5, 0.7).

| Recall | @IoU 0.3 | @IoU 0.5 | @IoU 0.7 |
|:---:|:---:|:---:|:---:|
| CoIn | 0.67 | 0.63 | 0.46 |
| + E3D | 0.84 | 0.79 | 0.61 |
| *Improvement* | 0.17 | 0.16 | 0.15 |

generation, a discrepancy exists with the annotated boxes. Consequently, this discrepancy results in a slightly higher increase in recall rate at lower IoU thresholds.

## 5 DISCUSSION AND CONCLUSION

This paper proposes a two-stage 3D object detection training strategy, E3D, exploring an approach to explicitly utilize the prior knowledge inherent in LMMs to enhance the capabilities of the sparsely-supervised 3D detectors. First, we develop a CSPT module to obtain accuracy seed points in point clouds by efficiently transferring high-confidence semantic masks extracted with LMMs. Next, we introduce a DCPG module to dynamically generate pseudo-label proposals within the multi-scale neighborhoods of seed points. Lastly, we propose a DS score as the criterion for NMS to select the high-quality pseudo-labels integrated with the CoIn training strategy to train the initial detector. After fine-tuning with sparsely annotated data, E3D demonstrated superior performance over the original sparsely-supervised 3D object detector on the KITTI dataset, and it maintained robust performance even as the amount of annotation decreased.

**Limitations.** One limitation is that the current E3D framework exhibits a relatively significant performance degradation when fine-tuning with annotation rates below 0.1%, which may result from the noise introduced by the extremely low annotations. Future efforts to explore efficient fine-tuning strategies to solve this problem.

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

APPENDIX

# A  THE VISUALIZATION OF THE EFFECT OF MASK SHRINK

The left side of Fig. 5 displays four scenarios of seed points (blue) directly using the prior information from the LMMs. As shown in the figure, pervasive noise exits in the seed points, which significantly hinders the subsequent generation of high-quality pseudo-labels. At the same time, we observe that the noise is primarily concentrated at the edges of the mask. Based on this finding, we design a mask shrink module based on boundary constraints. After incorporating this module, the effect on the seed points is shown on the right side of Fig. 5. It can be seen that we finally retained high-quality seed points.

Figure 5: Visualization of semantic seed points transformed from LMMs-extracted foreground mask. Direct transformation (left): Uncertainty edge segmentation, coupled with the inherent one-to-many nature of the pixel-to-point cloud, often results in a significant number of background points being mistakenly classified as foreground. Transformation with mask shrink (right): We only transfer the central region of the foreground mask onto the point cloud, which can eliminate edge semantic ambiguity and projection uncertainty.

## B  THE VISUALIZATION OF THE EFFECT OF DCPG

Fig. 6 upper and lower parts respectively showcase the bounding box pseudo-label fitting process for two instances. From these two examples, it can be seen that using a fixed parameter for the clustering radius $r$ makes it difficult to fit the most appropriate bounding box pseudo-labels. Moreover, combined with DS score and the NMS strategy, we subsequently filter out the low-quality pseudo-labels. Finally, it is the retained high-quality pseudo-labels that can support the training of a well-performing initial 3D detector.

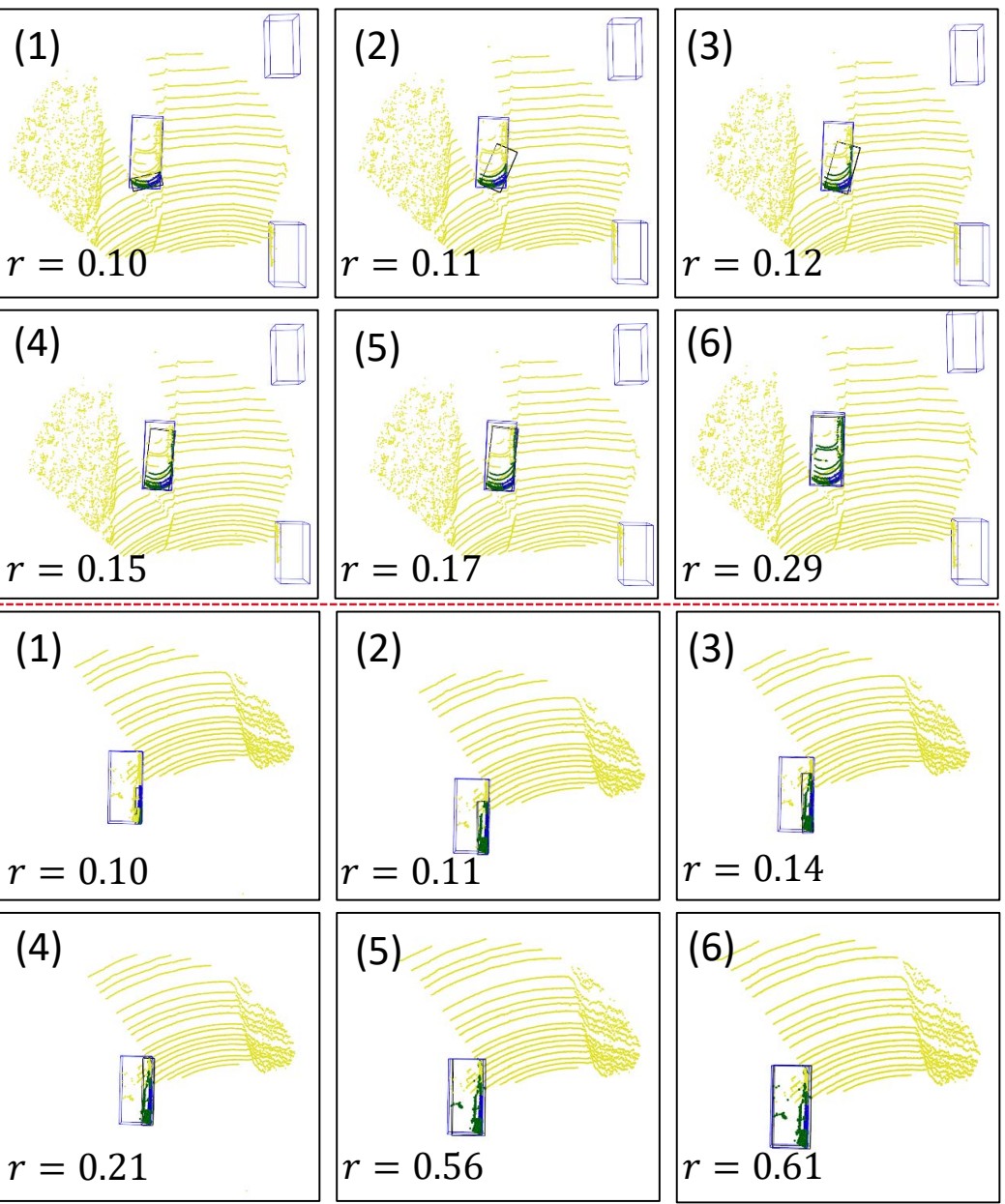

Figure 6: Visualization of the process of fitting bounding boxes with dynamic cluster radii in DCPG. By assigning different cluster radii $r$ to different seed points, our method is capable of capturing multi-scale foreground information, thereby fitting higher-quality pseudo-label proposals. Finally, we use the proposed DS score to rate each fitted bounding box, and in conjunction with NMS (Non-Maximum Suppression), only retain high-quality boxes as the final pseudo-labels.

## C    PSEUDO-LABEL QUALITY ASSESSMENT

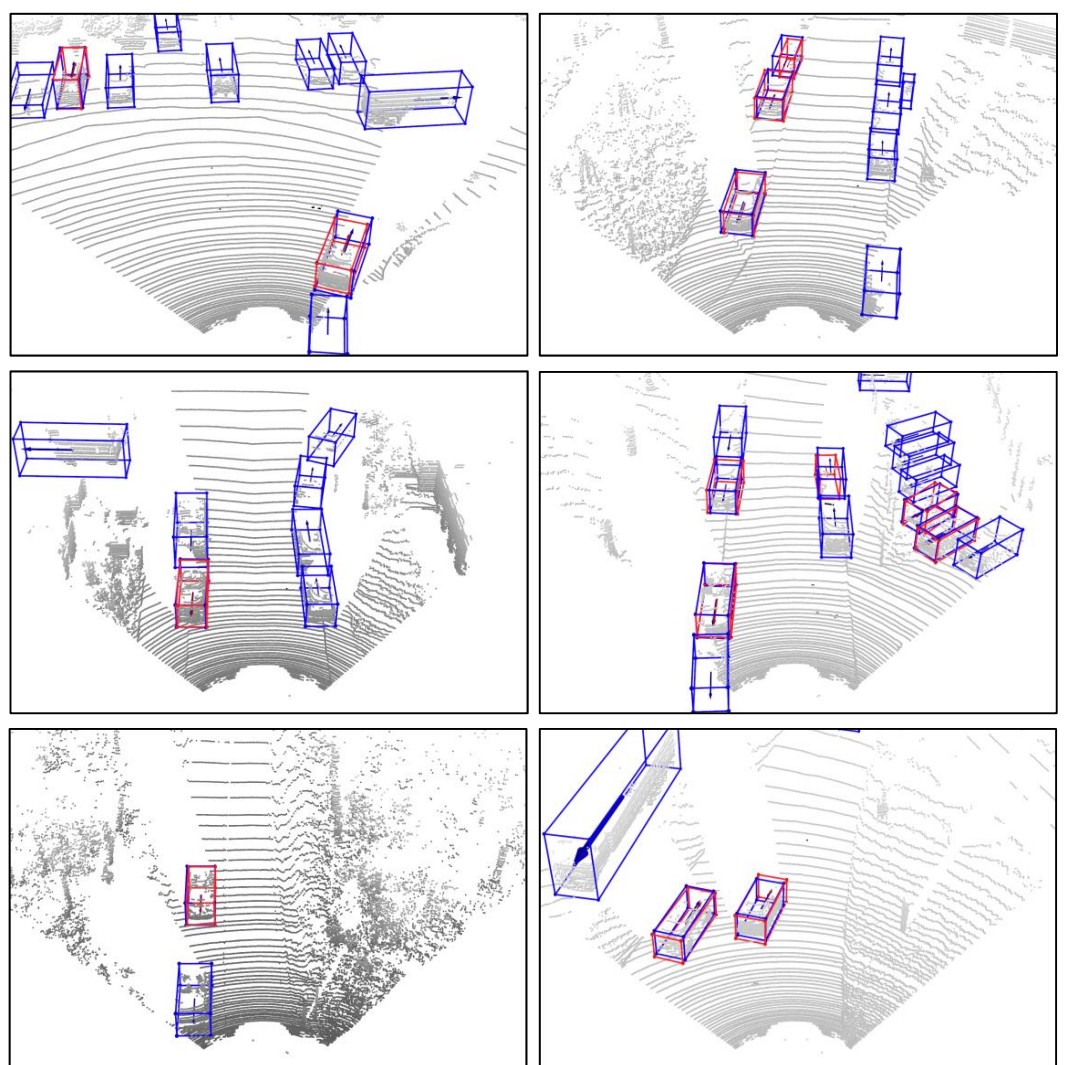

Figure 7: Visualization of pseudo-label quality assessment.

To visually demonstrate the E3D-generated pseudo-labels' quality, we simultaneously visualize them with the KITTI GT bounding boxes in Fig. 7. We represent the pseudo-labels generated by E3D with the red boxes and the GT annotations with the blue boxes. As shown in the figure, the red boxes exhibit characteristics close to the corresponding blue boxes in the majority of cases,

Table 7: Comparison of pseudo label quantities across different IoU thresholds.

|  | IoU$_{\leq 0.5}$ | $IoU_{<0.7,>0.5}$ | IoU$_{\geq 0.7}$ |
|---|---|---|---|
| Num. | 156 | 281 | 668 |
| Per. (%) | 14.12 | 25.43 | 60.45 |

indicating the high quality of the E3D-generated pseudo-labels. In addition, to quantitatively analyze the E3D-generated pseudo-labels' quality, we counted the number of pseudo-labels across various IoU thresholds, with the results recorded in Tab. 7. As demonstrated in the table, most of the generated pseudo-labels have an IoU with the GT above 0.7, and the ones with an IoU below 0.5 constitute only 14.12% of the total, which verifies the effectiveness of our proposed E3D.

