# OpenReview forum: "E3D: Enhancing Sparsely-Supervised 3D Object Detector with Large Multimodal Models"
_ICLR.cc/2025/Conference — ICLR 2025 Conference Withdrawn Submission_

### Official Review · Reviewer_htaw · 2024-10-22

**Soundness:** 2
**Presentation:** 1
**Contribution:** 1
**Rating:** 1
**Confidence:** 5

**Summary:**

This paper proposes a semi-supervised 3D detection framework that requires radar-vision paired data. The framework utilizes LMM and SAM to label the data, obtaining some 2D proposals. Then, the corresponding point cloud data is clustered to obtain 3D proposals and a semantic label is assigned by projecting points on images and exploiting LMM. Finally, empirical methods are used to refine the pseudo-labels to generate high-quality 3D pseudo-labels. These pseudo-labels are ultimately used to pre-train a point cloud 3D detector.

**Strengths:**

This paper proposes a system which can be used for generating 3D labels for training point cloud 3D detectors.

**Weaknesses:**

1. This paper uses heuristic methods to label Lidar-Image pairs, which serves as the initial annotation for pre-training the point cloud detector. This approach has little value in real-world industrial applications, as there are already many open-source image-point cloud 3D detection models with good performance available today. This reviewer encourages the authors  to refer to the survey "Multi-modal 3D Object Detection in Autonomous Driving: A Survey and Taxonomy" for more details. In addition, since this paper exploits Lidar-Image pairs to generate 3D pseudo-labels, the method you should compare against is a Lidar-vision fusion-based 3D detector, rather than point cloud based detector(Coin). However this paper completely overlooks all the work on Lidar-vision fusion-based 3D detectors.

**Questions:**

1. This paper is either unfinished or poorly written that it was difficult to understand. For example, in algorithm1, what are "shape[0]", "GroundRemove", "BoxFit".  I guess "GroundRemove", "BoxFit" are some heuristic operations including many empirical hyper-parameters. All these should be clearly presented so that the methods in the article can be reproduced.

2. The comparison between the pseudo-label from your method in stage one against the pseudo-label(or prediction results) from SOTA Lidar-vision fusion-based 3D detectors should be given.

---

### Official Review · Reviewer_JAy4 · 2024-10-29

**Soundness:** 2
**Presentation:** 3
**Contribution:** 3
**Rating:** 5
**Confidence:** 4

**Summary:**

This paper focuses on the sparsely-supervised 3D object detection and proposes a new method called E3D to improve the existing methods CoIn. By leveraging the rich priors provided by SAM and SemanticSAM, this paper can generate pseudo labels using the proposed CPST, DCPG, and DS score modules. With fine-tuning on sparse labels, it can finally obtain a high-performance 3D detector with much lower annotation costs. This paper evaluates the proposed method on the widely-used KITTI dataset, and the results show significant improvements when equipping the leading sparsely-supervised 3D object detector, i.e., CoIn, with E3D. The ablations and visualizations also prove the effectiveness of each module.

**Strengths:**

For writing quality and clarity:

- The paper is well organized, and the descriptions of the key motivation, ideas, and specific methods are straightforward, making it easy to reproduce and follow this work.
- The visualizations give intuitive explanations of the contribution of each module to the final results.

For key ideas:

- The idea of using the rich information from existing LMMs (e.g., SAM and SemanticSAM) to obtain pseudo labels and lower the annotation burden is good and sound, since the LMMs usually have great performance and generalization ability.
- The experimental results also show that the idea brings many improvements compared to the CoIn. This method only needs meager annotation costs (2%) to achieve the 90%+ performance of fully-supervised counterparts.

**Weaknesses:**

For writing:

- Minor typos: line 210, “which is except to”, I guess the authors mean “which is expect to”?

For related work:

- Lacks some discussion about weakly-supervised methods (e.g., VS3D[1], WS3D[2], ViT-WSS3D[3]) in the related work section.

For method designs:

- Some processes need further explanations. Please see the questions.
- The proposed DS score module seems very dataset specific, which may hinder the generalization ability of the whole method. Please see the questions.

For experiments:

- At the current stage, KITTI is somehow old and not a large-scale dataset, and the diversity of samples is also not good. More experiments on modern datasets (e.g., nuScenes and Waymo) are needed.
- Some statistics are needed to get a more comprehensive evaluation of the superiority of this method. Please see the questions.
- It needs a failure case study.
- Some experiments are missing to support some claims: In the abstract, the authors say, “we have verified our E3D in the zero-shot setting.” However, it seems there is no corresponding experiment.

[1] Qin, Zengyi, Jinglu Wang, and Yan Lu. "Weakly supervised 3d object detection from point clouds." *Proceedings of the 28th ACM International Conference on Multimedia*. 2020.

[2] Meng, Qinghao, et al. "Weakly supervised 3d object detection from lidar point cloud." *European Conference on computer vision*. Cham: Springer International Publishing, 2020.

[3] Zhang, Dingyuan, et al. "A simple vision transformer for weakly semi-supervised 3d object detection." *Proceedings of the IEEE/CVF International Conference on Computer Vision*. 2023.

**Questions:**

For method designs:

- In the dynamic cluster pseudo-label generation, this paper updates the radius linearly (E.q. (4)). The variable in this equation is $t$, which is the index of points in the seed points set; what is the physical meaning of this? Why does the order of the points in the set determine the radius?
- For the distribution constraint score, the method claims the distances from interior points to the proposal boundary follow a Gaussian distribution N(0.8; 0.2). Is this claim also suitable for other datasets and any type of object? I do not think the points from cars and pedestrians have the same distribution.
- For the meta-shape constraint score, the method uses the mean shape of each category as the template. This will have two main issues:
    - We need to adjust it to adapt to different datasets.
    - This will not work well if a category with instances of diverse shapes (e.g., trucks) exists. Consequently, the pseudo-labels generated by this method will face diversity problems.

For experiments:

- It would be better to provide some results on nuScenes or Waymo.
- It would be better to list the results of Ped. and Cyc. in Tab. 2-5.
- It would be better if the authors conducted a failure case study, which would help us understand the limits of the proposed method and get further improvements.
- According to the visualization in Fig. 7, the pseudo-labels seem to have low recall:
    - Why could this happen?
    - It would be better to provide the mAP of pseudo labels.
- For comparison with different annotation rates, I am personally curious about the results if we use E3D with a 100% annotation rate. Since we use the priors from LMMs, can we perform better than the fully-supervised baseline?

---

### Official Review · Reviewer_2iEe · 2024-11-02

**Soundness:** 2
**Presentation:** 3
**Contribution:** 2
**Rating:** 5
**Confidence:** 4

**Summary:**

The paper is about label-efficient 3D object detection, which is an important topic in autonomous driving . The novelty is supposed to be the usage of LMMs which DCPG and DS score follow in generating reliable pseudo-label. The method is supposed to be universal for any point-based 3D object detection.

**Strengths:**

The topic of the paper is vital to autonomous driving since the 3D annotation cost is huge. The proposed method brings the recent progress in foundation models to tackle this problem, which is believed to be universal. In general, the paper is well understood.

**Weaknesses:**

The novelty of the paper is limited in its excessive steps despite the usage of LMM, which renders the tuning of the methods as a delicate job and further requires more comprehensive experiments. In addition, there are many related references to be compared with to enhance its  superiority.

**Questions:**

I have several concerns as follow
1. Some parts are a bit misleading. For example, the latter part of section 2.3 is mainly about the introduction of MixSup. What does SAL mean? 'Text prompt' at line 203 is confusing at first glance and 'Except' at line 210 seems to be a typo.
2. LMMs-guide semantic extraction is very similar to PointSAM in MixSup. Considering the usage of LMM is supposed to be the core contribution. Their comprehensive comparison is expected.
3. There are many works as a preprocessor to generate pseudo-labels. For example, OC3D: Weakly Supervised Outdoor 3D Object Detection with Only Coarse Click Annotation in arxiv,  and Commonsense Prototype for Outdoor Unsupervised 3D Object Detection in CVPR 2024. They may differ in the way of supervision but with the same goal to generate pseudo-labels. Thus the superiority of the proposed method to these works are better validated.
4. In terms of experiments, the performance on other benchmarks like WOT is better presented for a convincing conclusion. In addition, the combination of the proposed method with other 3D object detection besides CoIn is also encouraged to prove its universality.
5. The tuning of the proposed method seems to be non-trivial since there are many parameters in 3 components. Can the authors elaborate more on this? For example, confident points filtering may also encounter semantic ambiguity around object edge according to mask shrink in Eq.3.

---

### Official Review · Reviewer_Zb61 · 2024-11-03

**Soundness:** 3
**Presentation:** 3
**Contribution:** 3
**Rating:** 6
**Confidence:** 4

**Summary:**

This paper proposes a new method for sparsely supervised 3D object detection, E3D, which enhances the feature discrimination ability of sparsely supervised 3D detectors by utilizing the 2D semantics generated by large-scale multimodal models.

There are three core modules in this paper: Confident Points Semantic Transfer extracts seed points from 2D semantic information generated by large-scale multimodal models (LMMs). Dynamic Cluster Pseudo-label Generation generates pseudo-labels from multi-scale neighborhood information. The Distribution Shape score is used to screen high-quality pseudo-labels unsupervised.

On the KITTI dataset, E3D performs better under different sparse annotation rates.

**Strengths:**

1. The semantic information of 2D images is obtained and transferred through LMMs, which makes up for the disadvantage of insufficient annotation of 3D point cloud data
2. The results of the experiment are remarkable, especially for more difficult categories, such as pedestrians and bicycles.

**Weaknesses:**

1. Perform relevant experiments only on the KITTI dataset, and hope to try to train and test on a larger dataset, such as the Waymo dataset
2. The model training process uses large-scale multimodal models and DBSCAN clustering algorithms, which will introduce large computational overhead. However, this article does not provide a comparison of the cost with relevant models.
3. There is a lack of experimental verification of the necessity of introducing two large-scale multimodal models, FastSAM and Semantic SAM. Lack of experimental validation of the importance of the various mass fractions in pseudo-label screening methods. The baseline method is not specified in Table 5.

**Questions:**

1. Please supplement the necessary ablation experiments to verify the effectiveness of the modules presented in this paper
2. The radar perception range is often larger than that of the foreground image, and when the full-view image cannot be obtained, will the E3D method proposed in this paper be significantly affected?

---

### Official Review · Reviewer_wkAx · 2024-11-04

**Soundness:** 3
**Presentation:** 3
**Contribution:** 2
**Rating:** 5
**Confidence:** 4

**Summary:**

This paper proposed a new framework to generate pseudo-3D boxes for 3D object detectors. It mainly contains three steps. Firstly, seed points are obtained via 2D LMM models by the projection to 3D. Then, pseudo labels are generated by these seeds to train detectors. Finally, a filtering strategy is applied to select high-quality pseudo labels to refine the model. Experiments on the KITTI are conducted and show impressive improvement with 2% manually annotated labels.

**Strengths:**

1. The paper is well-written and easy to follow. The method and implementation details are clear. Hyper-parameters are also provided. Therefore, the results should be easy to reproduce.
2. The author conducted many ablation studies to reveal each component's effectiveness. These results help readers to grasp all the details of the method.
3. The method demonstrates impressive overall performance on the KITTI dataset.

**Weaknesses:**

1. The technical novelty is limited. Even though each component is very reasonable, they are already developed in previous methods or are very heuristic and rely on carefully tuned hyper-parameters. The paper does provide many in-depth insights. Please consider providing more discussion with previous label efficient 3D object detection methods, especially those also using 2D cues to lift them into 3D pseudo-labels, such as the following suggestions [a-d].
2. The experiments are only conducted on the KITTI dataset. Therefore, the generalization ability is not explicitly proved. Please consider providing more experimental analysis on more datasets, such as the NuScenes, Waymo, and H3D.

- [a] Weakm3d: Towards weakly supervised monocular 3d object detection
- [b] Weakly supervised monocular 3d object detection using multi-view projection and direction consistency
- [c] General Geometry-aware Weakly Supervised 3D Object Detection
- [d] Transferable Semi-Supervised 3D Object Detection From RGB-D Data

**Questions:**

Please refer to the weaknesses for major concerns. Besides, there are some minor questions:
1. The formulation in the introduction section can be further improved. For example, in lines 77-80, the authors proposed two questions. However, the question is general and does not specifically correspond to the method solutions.
2. Why not consider also using the ground truth labels in the first stage?

---

### Note · Authors · 2024-11-14

I have read and agree with the venue's withdrawal policy on behalf of myself and my co-authors.